# Enabling nonconjugated polyesters emit full-spectrum fluorescence from blue to near-infrared

Bo Chu[1], Xiong Liu[1,2,3], Zuping Xiong[1,2,3], Ziteng Zhang[1,2,3], Bin Liu[4], Chengjian Zhang [1], Jing Zhi Sun [1,3], Qing Yang [5], Haoke Zhang [1,2,3] ✉, Ben Zhong Tang [1,6] ✉ & Xing-Hong Zhang [1] ✉

Near-infrared luminophores have many advantages in advanced applications, especially for structures without π-conjugation aromatic rings. However, the fabrication of red clusteroluminogens from nonconjugated polymers is still a big challenge, let alone the near-infrared clusteroluminogens. Here, we develop nonconjugated luminophores with full-spectrum from blue to near-infrared light (470 ~ 780 nm), based on color phenomenon of nonconjugated polyesters synthesized from the amine-initiated copolymerization of epoxides and cyclic anhydrides. We reveal that amines act as initiators attached to polymer chain ends. The formation of various amine-ester complexes in polyesters induces red to near-infrared light, conceptually, amine-ester complexed clusteroluminescence via intra/inter-chain charge transfer. Significantly, emission colors can be easily tuned by the contents and types of amines, microstructures of polyesters, and their concentration. This work provides a low-cost, scalable platform and strategy for the production of high-efficiency, multicolor luminescent materials.

Light, as a magnificent and fantastic magician, lights up the colorful world, especially the emerging luminescent materials that play an essential role in the remarkable advancement of modern civilization. Classic artificial luminophores are organic compounds with π-conjugated structures, widely used in interdisciplinary fields such as illumination[1–4], bioimaging[5–8], and sensor[9–11] owing to high efficiency and multicolor luminescence. Over the past decades, through-bond conjugation[12–14] has been the prerequisite to guide the design of luminophores by increasing π-conjugated structures. However, conventional π-conjugated molecules suffer from quenching effects at aggregate states, poor processability, complicated synthesis, and environmental hazards[12,15–17], which limits practical applications.

In recent years, many nonconjugated polymers, including natural and synthetic heteroatom-rich (e.g., O, N, and S) polymers[18–22] are observed to show abnormal luminescence, namely clusterolumines-cence (CL)[23–25] manipulated by through-space interactions (TSIs)[26,27]. These nonconjugated heteroatom-rich polymers have the advantages of easy processing, biocompatibility, etc., and can be mass-produced from renewable, low-cost biomass resources. Nevertheless, lacking universal CL theory to guide the structural design of luminophores,

[1]National Key Laboratory of Biobased Transportation Fuel Technology, International Research Center for X Polymers, Department of Polymer Science and Engineering, Zhejiang University, Hangzhou 310058, China. [2]Zhejiang-Israel Joint Laboratory of Self-Assembling Functional Materials, ZJU-Hangzhou Global Scientific and Technological Innovation Center, Zhejiang University, Hangzhou 311215, China. [3]Centre of Healthcare Materials, Shaoxing Institute, Zhejiang University, Shaoxing 312000, China. [4]School of Energy and Power Engineering, North University of China, Taiyuan 030051, P. R. China. [5]State Key Laboratory of Silicon Materials, Zhejiang University, Hangzhou 310027, China. [6]School of Science and Engineering, Shenzhen Institute of Aggregate Science and Technology, The Chinese University of Hong Kong, Shenzhen (CUHK-Shenzhen), Guangdong 518172, China. ✉e-mail: zhanghaoke@zju.edu.cn; tangbenz@cuhk.edu.cn; xhzhang@zju.edu.cn

which makes photophysical performances difficult to regulate only with short-wavelength emission (400–500 nm, few at 600 nm) and low quantum yields (QYs)[19,21,28–32]. Our previous works have disclosed that QYs and wavelengths of ester clusters in polyesters could be regulated by hierarchical structures, which achieved white-to-orange lights[33,34]. Hence, it is highly desired but a big challenge to develop high-efficiency, red, and even near-infrared (NIR) nonconjugated polymer materials.

Accidentally, our lab observed that nonconjugated polyesters from the copolymerization of epoxides and cyclic anhydrides catalyzed by organic amine-alkyl boron Lewis pairs[35] tend to be abnormal dark color and emit red lights under UV light. The intrinsic mechanism of such amazing stable colors and red luminescence was unclear. Herein, we revealed that amines acted as initiators attached to polymer chain ends and formed stable amine-ester complexes, which induce red-to-NIR CL at 600–780 nm through intra-chain/inter-chain charge transfer. Significantly, we proposed an engineering strategy of amine-ester complexes by the initiation of organic amines to transform large-scale, low-cost polyesters into blue-to-NIR luminophores. Meanwhile, CL colors can be easily tuned from blue to NIR by concentration, types of amines, and chain structures of polyesters.

## Results

### Synthesis and characterization of polyesters

The polyesters **P1** and **P2** were obtained by the copolymerization of epoxides and cyclic anhydrides, which were catalyzed by heterogeneous inorganic zinc-cobalt (III) double metal cyanide complex (Zn–Co$^{III}$ DMCC)[36] as the control group with no interference of Zn-OH on luminescence[33,34] (Supplementary Figs. 1-2). Meanwhile, the polymers of **P1-aTEA/DBU/mTBD/TBD** and **P2-aTEA/DBU/mTBD/TBD** catalyzed by organic Lewis pairs[35] of triethylborane (TEB) with different amine, triethylamine (TEA), 1,8-diazabicyclo[5.4.0]undec-7-ene (DBU), 7-methyl-1,5,7-triazabicyclo[4.4.0]dec-5-ene (mTBD) and 1,5,7-triazabicyclo[4.4.0]dec-5-ene (TBD), were obtained by using different molar ratios (a%) of amines from 0.5% to 5.0% (Supplementary Figs. 3–6). Then, the chemical structures of all the polyesters were well confirmed, showing that the amines attack the propylene oxide first and attach to polymer chain ends (Supplementary Figs. 7–67), which discloses the initiation mechanisms of amines and paves the way for studying the origin of dark color in amine-initiated polyesters.

### Photophysical properties

The photophysical properties of **P1** with only ester groups were systematically characterized by UV–Vis and photoluminescence (PL) spectra. Dominant absorption bands (250–300 nm) and maximum PL peaks (at ~465 nm) indicated typical $(n,\pi^*)$ transition of isolated ester groups[33,34,37], which is clearly enhanced by the increased concentration ($c$) from $10^{-5}$ M to $10^{-1}$ M (Supplementary Figs. 68–70). The above results suggest that **P1** only enhances the intrinsic $(n,\pi^*)$ transition of esters groups with weak TSIs. Then, the color origin in **P1-aTEA** was explored. Firstly, two control experiments of **P1-2.0TEA** catalyzed by TEA and TEB-TEA, respectively, showed nearly the same PL spectra (Supplementary Figs. 71–73), which completely excludes the influence of TEB on CL properties, indicating the dominant role of TEA in coloration. As shown in Supplementary Figs. 74–81, in DCM solution, with a% of TEA increasing from 0.5% to 5.0% or $c$ increasing from $10^{-5}$ to $10^{-1}$ M, the absorption band of **P1-aTEA** at 300–600 nm appeared and was enhanced, suggesting that complexation between amines and esters induced the new strong TSIs.

To further disclose the origin of new strong TSIs, the PL spectra of **P1-aTEA** were further investigated under different a% of TEA. In Fig. 1A, besides the short-wavelength blue emission corresponding to the $(n,\pi^*)$ transition of ester groups similar to **P1, P1-5.0TEA** exhibited an additional long-wavelength PL peak at ~600 nm. It is noteworthy that the ratio of the long-wavelength CL (600 nm) intensity versus the short

one (474 nm) increased with the increasing a% of TEA (Fig. 1B, C), suggesting enhancing TSIs between amines and esters. Hence, the fluorescence photographs for **P1-aTEA** under 365 nm UV-light exhibited significant color change from blue to white, orange, and red when the a% increased from 0.0 to 5.0% (Fig. 1D). Among all the polymers, **P1-5.0TEA** shows the highest solid-state QY of 15.8% (Supplementary Table 1). Meanwhile, pronounced excitation-dependent emission (EDE) effect from the strong peak at 470 nm to long-wavelength CL at ~600 in **P1-0.5TEA** indicates relatively weak TSIs, but as a% further increased to 5.0%, the EDE effect was gradually weakened and long-wavelength CL at ~600 nm became dominant (Supplementary Figs. 82–96), which also suggest that increasing the molar ratio of TEA can gain strong TSIs. PL decay curves show that the lifetime of the short-wavelength and long-wavelength emissions are both in the range of 1–10 ns with a slight difference, indicating the nature of fluorescence (Supplementary Figs. 97–110). Further, concentration-dependent PL spectra of **P1-aTEA** in DCM showed similar CL intensity enhancement on the formation of complexes with $c$ increasing from $10^{-5}$ M to $10^{-1}$ M (Supplementary Figs. 111–117). These results suggest that the complexes-induced TSIs were formed once the TEA was utilized in the polymerization, and the incremental molar ratio of TEA or solution concentration only increases the number of amine-esters complexes without changing the chemical structures of complexes.

According to our previous reports[33], **P1** shows a typically loose helix conformation so that the involved TSIs are very weak and result in blue emission from isolated ester groups. However, the dense folding structure of **P2** can induce strong TSIs among ester groups and produce orange CL with a maximum emission wavelength ($\lambda_{em}$) of 570 nm. Inspired by this, we can make stronger complexes by introducing TEA into **P2** to generate redshifted CL. The obtained **P2-aTEA** showed a stronger absorption band at 400–600 in UV–Vis spectra relative to **P1-aTEA** under equal a% (Supplementary Figs. 118–124); meanwhile, the absorption band had a dramatic concentration-enhanced effect with the concentration increasing from $10^{-5}$ M to $10^{-1}$ M, which suggests that denser folding structures did favor to the formation of a strong complex. In Fig. 2A, B, Supplementary Figs. 125–138 and Supplementary Table 2, under 365 nm excitation, the PL spectra of **P2-aTEA** in both solid and concentrated DCM solution ($c = 10^{-1}$ M) suggest that the short-wavelength emission (470–490 nm) keeps almost no change, but the long-wavelength CL continuously shifts from 600 nm to 660 nm when a% of TEA increased from 0.5% to 5.0%. Moreover, high a % in **P2-aTEA** induced another new redder CL at 740–750 nm in the NIR region. The plots of PL intensity versus a% further exhibited that blue PL dropped rapidly and NIR CL was intensified steadily with increasing of a% (Fig. 2C). PL decay curves also showed that the lifetimes of all these emissions were from fluorescence (Supplementary Figs. 139–152). In Fig. 2D, the phenomena of excitation-independent emission on red and NIR CL of **P2-5.0TEA** in concentrated DCM solution ($c = 10^{-1}$ M) demonstrated the stabilized clusters of amine-ester complexes.

To further investigate the formation of NIR-CL clusters, concentration-dependent PL spectra of **P2-aTEA** (a% is from 0.5% to 5.0%) in DCM solution with $c$ from $10^{-5}$ M to $10^{-1}$ M were measured, which shows similar properties (Supplementary Figs. 153–159). Hence, taking **P2-5.0TEA** with the strongest NIR CL as an example, it exhibited full-spectra emission from blue to NIR only by changing the concentration (Fig. 2E). The plots of relative PL intensity versus $c$ for each emission (blue PL at 470 nm, red CL at 600–660 nm and NIR CL at 740 nm) indicate that there are four stages for the variation of relative PL intensities, which shows three turning points $c_1$ ($10^{-4}$ M), $c_2$ ($10^{-3}$ M), and $c_3$ ($10^{-2}$ M) corresponding to the critical cluster concentration (CCC) of red CL, the concentration at which more esters participate in the complexation and the CCC of NIR CL, respectively. (i) In the first stage below $c_1$, blue PL from isolated esters was dominant with no amine-ester complexes, suggesting a weak TSI in such a dilute solution.

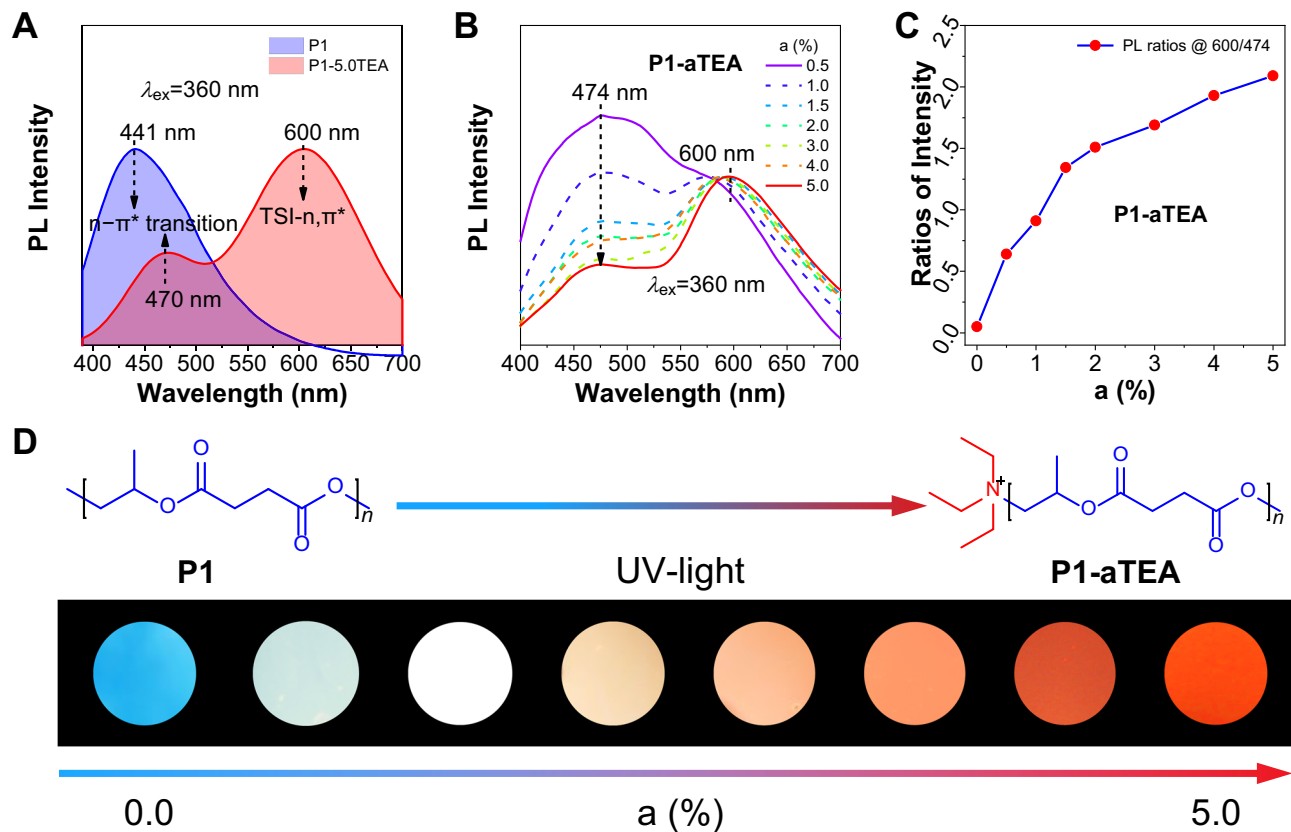

**Fig. 1 | Photophysical properties and fluorescence photographs of P1 and P1-aTEA with different molar ratios (a%) of TEA from 0.5% to 5.0%. A** Normalized photoluminescence (PL) spectra of **P1** and **P2-5.0TEA** in solid at $\lambda_{ex} = 360$ nm. **B** PL spectra of **P1-aTEA** in solid at $\lambda_{ex} = 360$ nm. **C** Plots of intensities ratios versus contents of TEA for **P1-aTEA** in solid. The intensity ratio is the emissions of 600 nm/474 nm. $\lambda_{ex} = 360$ nm. **D** Photographs and chemical structures of **P1-aTEA** in solid films taken under 365 nm UV light.

(ii) In the range of $c_1$ to $c_2$, the complexes began to form, and the red CL was induced and enhanced with further increasing of $c$. (iii) In the third stage, once $c$ reached the $c_2$, more isolated esters began to disappear and involve in complexation, causing the decreased relative intensity of the blue PL, then more and more esters involved in complexation with the increased $c$. (iv) When the $c$ reached $c_3$, saturated clusters of red CL formed, and the increasing $c$ leads to long-range complexes owing to more isolated esters participating in complexation. Then, the long-range complexes started to induce and enhance NIR CL, further weakening the blue PL. The above results not only visualized the clusterization process of NIR CL but also disclosed the existence of long-range complexes between TEA and multiple esters. Finally, clusteroluminogens (CLgens) with white, orange, and red emission colors were obtained (Fig. 2G). The NIR CL in amine-initiated polyesters suggests the efficient role of complexation in generating strong TSI.

## Amine-ester complexes

Extensive characterization and theoretical calculations were carried out to disclose the complexation and luminescent mechanism of amine-ester complexes. Firstly, the PL spectra of **P1-2.0amine** initiated by four different organic amines (TEA, DBU, mTBD, and TBD) were measured, and all exhibited the same CL peaks (Fig. 3A and Supplementary Fig. 160), and so were **P2-2.0amine** (Supplementary Figs. 161–162), indicating that nitrogen on amines plays an important role in red-to-NIR CL. Meanwhile, **P1-2.0DBU** had the strongest red CL, and **P2-2.0DBU** also had the strongest red CL and NIR CL. Interestingly, when the molar ratio a% of DBU is increased to 5.0%, **P1-5.0DBU** also displayed obvious NIR CL at 740 nm, which is absent in **P1-5.0TEA** (Fig. 3B), suggesting that DBU can form strong amine-ester complexes for NIR CL. Then, time-dependent NMR spectra of various mixtures of

DBU and ester-containing structures were measured to reveal the complexation process and mechanism, where ester-containing structures include polymers (**P1** and **P2**) and the corresponding small molecules (dimethyl succinate (DS), dimethyl citraconate (DC), succinic anhydride (SA) and citraconic anhydride (CA)). In Fig. 3C, D, time-dependent $^1$H- and $^{13}$C-NMR spectra of **P1@DBU** (@ represents the mixing process) with 5:1 molar ratio of structural units of **P1** to DBU show that the major changes of atomic chemical shift ($\Delta\delta_H$ for hydrogen atoms, $\Delta\delta_C$ for carbon atoms) are 2–5 which are adjacent to two different nitrogen atoms on DBU. To confirm which nitrogen atom on DBU has dominant electronic interaction with esters, the plots of the absolute value of chemical shift changes ($\Delta\delta_H$, $\Delta\delta_C$) versus atom positions on DBU for **P1@DBU** were investigated (Fig. 3E). The maximum $\Delta\delta_H$ and $\Delta\delta_C$ were located at positions 4 or 5, while positions 2 and 3 had only negligible change. Moreover, NMR spectra of **P2@DBU, DS@DBU, DC@DBU, CA@DBU**, and **SA@DBU** all displayed the same trend (Fig. 3F and Supplementary Figs. 163-176), indicating that the nitrogen atom closed to positions 4 and 5 on DBU forms complexes with ester units. Especially, NMR spectra of **SA@DBU** with various molar ratios of SA to DBU from 0:5 to 5:0 disclosed the molar complexation ratios of DBU to SA is 2:1 (Supplementary Figs. 177–197). When TEA was mixed with ester-containing structures (Supplementary Figs. 198–205), smaller $\Delta\delta_H$ and $\Delta\delta_C$ of methylene and methyl on TEA relative to DBU were also observed, which also confirmed the structures of amine-ester complexes.

We next disclosed the electronic nature of amine-ester complexes in polyesters via theoretical calculation. Based on a single polymer chain with three repeating units, theoretical calculations were carried out to optimize the excited-state geometries of **P1@DBU, P2@DBU, P1-DBU**, and **P2-DBU** respectively (Fig. 3G and Supplementary

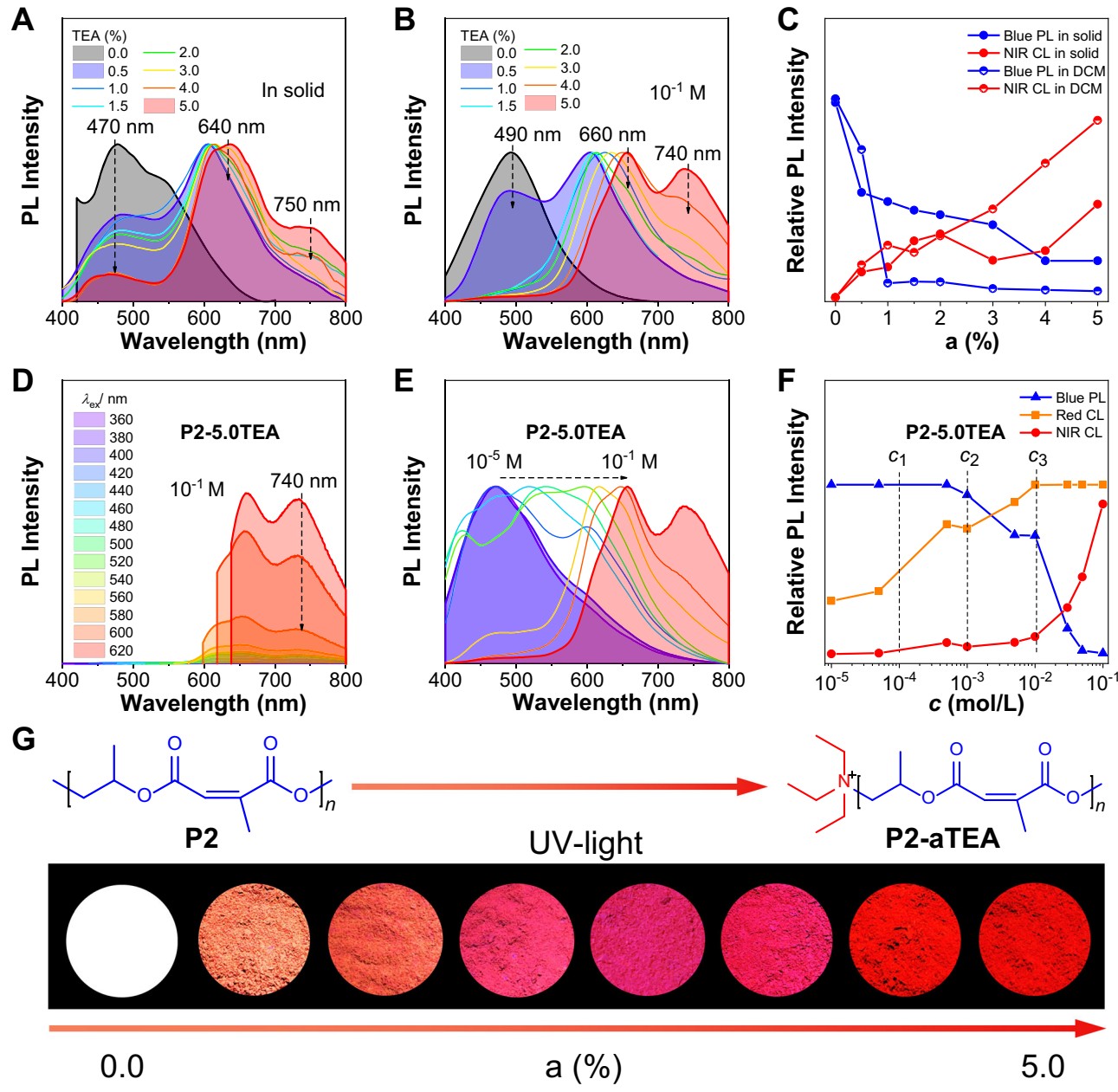

**Fig. 2 | Photophysical properties and fluorescence photographs of P2 and P2-aTEA with different molar ratios (a%) of TEA from 0.5% to 5.0%. A**, **B** Normalized PL spectra of **P2-aTEA** in solid and DCM solution, respectively, at $\lambda_{ex}$ = 360 nm. The concentration of **P2-aTEA** in DCM is $10^{-1}$ M. **C** Plots of relative PL intensities of blue PL/NIR CL versus contents of TEA for **P2-aTEA** in both solid and DCM of $10^{-1}$ M at $\lambda_{ex}$ = 360 nm. **D** PL spectra of **P2-5.0TEA** in DCM solution of $10^{-1}$ M. **E** Concentration-dependent PL spectra of **P2-5.0TEA** in DCM at $\lambda_{ex}$ = 360 nm. Concentration is from $10^{-5}$ to $10^{-1}$ M. **F** Plots of relative PL intensities of blue PL, red CL, and NIR CL versus contents of TEA for **P2-5.0TEA** in DCM of $10^{-1}$ M at $\lambda_{ex}$ = 360 nm, respectively. Concentration is from $10^{-5}$ to $10^{-1}$ M. **G** Fluorescence photographs and chemical structures of **P2-aTEA** in solid powders taken under 365 nm UV light.

Fig. 206). For **P1@DBU**, the highest occupied molecular orbital (HOMO) and lowest unoccupied molecular orbital (LUMO) were located on nitrogen atoms of DBU and esters of polyesters, respectively. For DBU-initiated **P1-DBU**, the HOMO and LUMO were located on ester moieties and the quaternary nitrogen atoms of terminal DBU, respectively. These results suggested that all the electron transitions occur between DBU and ester moieties. Meanwhile, the hole (red color) and electron (blue color) of optimized excited-state geometries of **P1@DBU**, **P2@DBU**, **P1-DBU**, and **P2-DBU** were further simulated (Fig. 3H and Supplementary Fig. 207). The low-lying excited state with the intra/interchain charge transfer or through-space charge transfer (TSCT) effect[38,39] was mapped for both **P1@DBU** and **P1-DBU**, further confirming the chemical and electronic structures of amine-ester

complexes. Moreover, the physical mixing or chemical bonding methods demonstrated the existence of both interchain and intra-chain amine-ester complexes and TSIs.

Besides, the origins of CL from amine-ester complexes were further investigated based on model small molecules. In Fig. 4A and Supplementary Fig. 208, the PL spectra of pure DC and DS only showed optimal PL peaks at 420 ~ 450 nm corresponding to the intrinsic $(n,\pi^*)$ transition of ester groups. Once TEA and DBU with molar ratios of 1/640 were added into bulk DC or DS, obvious redshifted CL at 500 ~ 605 nm appeared for **DS@DBU**, **DC@TEA**, and **DC@DBU**, indicating the formation of stable complexes. Moreover, except for **DS@TEA**, time-dependent PL intensities for **DS@DBU**, **DC@TEA**, **DC@DBU** clearly displayed that the long-wavelength CL at

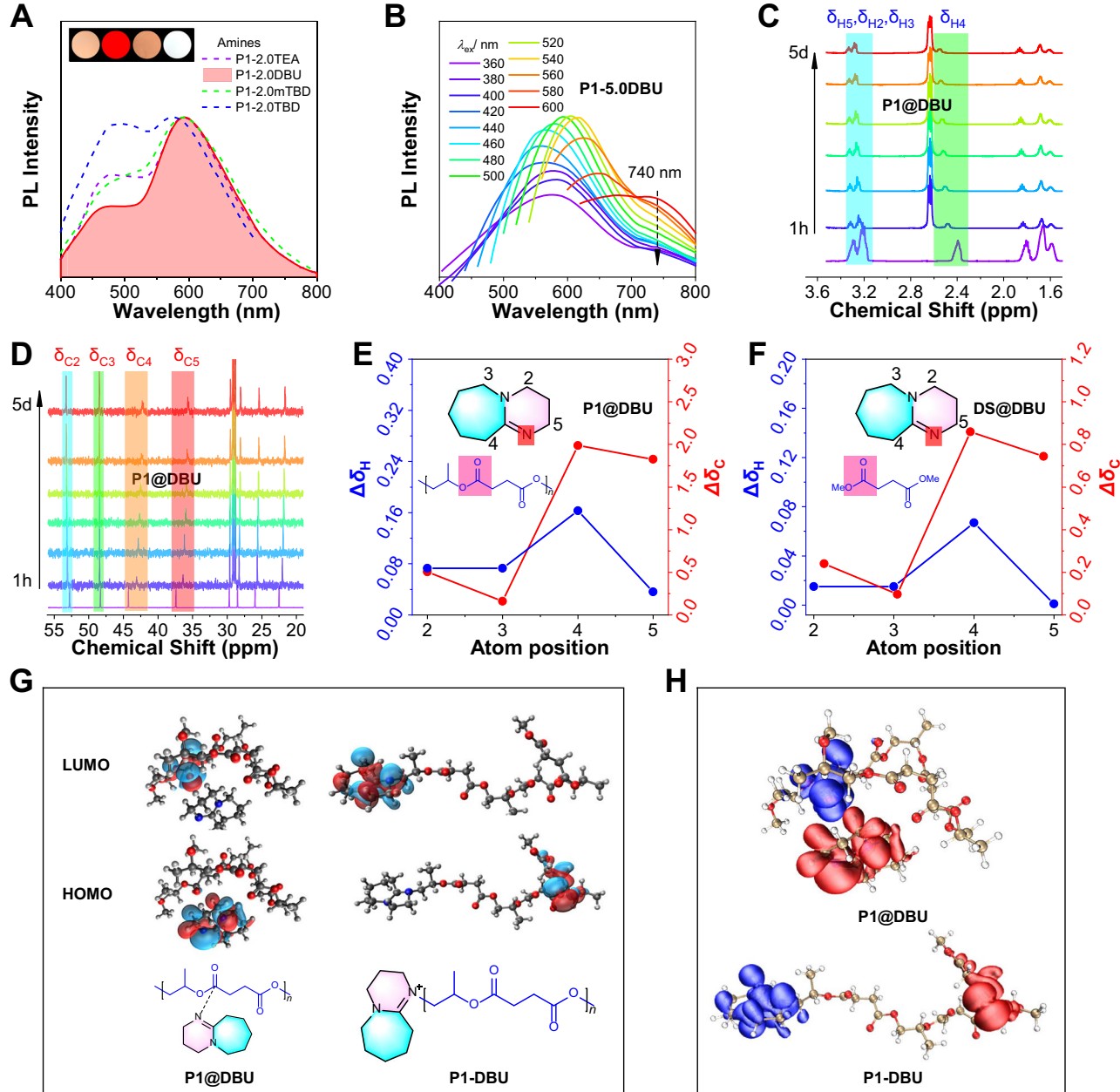

**Fig. 3 | Photophysical properties, dynamical NMR characterization, and theoretical calculation of amine-ester complexes. A** Normalized PL spectra of **P1-2.0TEA**, **P1-2.0DBU**, **P1-2.0mTBD**, and **P1-2.0TBD** in solid at $\lambda_{ex}$ = 360 nm. Inset: photographs are taken under 365 nm UV light of **P1-2.0TEA**, **P1-2.0DBU**, **P1-2.0mTBD**, and **P1-2.0TBD** from left to right. **B** PL spectra of **P1-5.0DBU** in the solid state were recorded at different excitation wavelengths. **C** Time-dependent $^1$H-NMR and **D** $^{13}$C-NMR spectra of **P1@DBU** were obtained from mixing of **P1** and DBU. The molar ratio of structural units of **P1** to DBU is 5:1. @ represents the symbol of the mixing process. The h and d represent hours and days, respectively. **E, F** Plots of the absolute value of chemical shift changes ($\Delta\delta_H$ for hydrogen atoms, $\Delta\delta_C$ for carbon atoms) versus atom positions on DBU for **P1@DBU** and **DS@DBU**, respectively. **DS** is the model compound of **P1**. The molar ratio of DS to DBU is 5:1. **G** Frontier molecular orbitals and **H** Hole (red color) and electron (blue color) of optimized excited-state geometries of **P1@DBU** and **P1-DBU** calculated by TD-DFT method at B3LYP-D3/6–31 G(d,p) level, Gaussian 09 program. HOMO: the highest occupied molecular orbital, LUMO: the lowest unoccupied molecular orbital.

500–605 nm enhanced with increasing treatment time (Fig. 4B and Supplementary Figs. 209–212), and the complexation rate of **DC@DBU** is the fastest with the reddest CL at 605 nm, which further proves that the red CL in **P1-amine** and **P2-amine** originates from amine-ester complexes. Meanwhile, increasing molar ratios of TEA or DBU relative to DC or DC from 1/640 to 1/20 also shifted the CL to the long-wavelength range, indicating the process of dynamic complexation (Supplementary Figs. 213–216).

However, the above results cannot clarify the origin of NIR CL. Therefore, temperature-dependent PL spectra of **DC@DBU** in bulk were recorded to verify the long-range amine-ester complexes

corresponding to the NIR CL. In Fig. 4C, D and Supplementary Figs. 217–218, there were two opposite stages for the variation of CL intensity for red CL ($\lambda_{em}$ = 640 nm) and NIR CL ($\lambda_{em}$ = 720 nm). At the first stage, **DC@DBU** showed original red CL at 640 nm based on amine-ester complexes under 77 K, and then the red CL was enhanced with the temperature increasing from 77 K to 177 K. Meanwhile, NIR CL at 700–800 nm gradually appeared and dominated leading to broadened spectra, because more molecules started to unfreeze and collide with each other to produce long-range complexes. In the second stage, as the temperature further increased from 177 K to 297 K, molecular collisions intensified, and long-range complexes

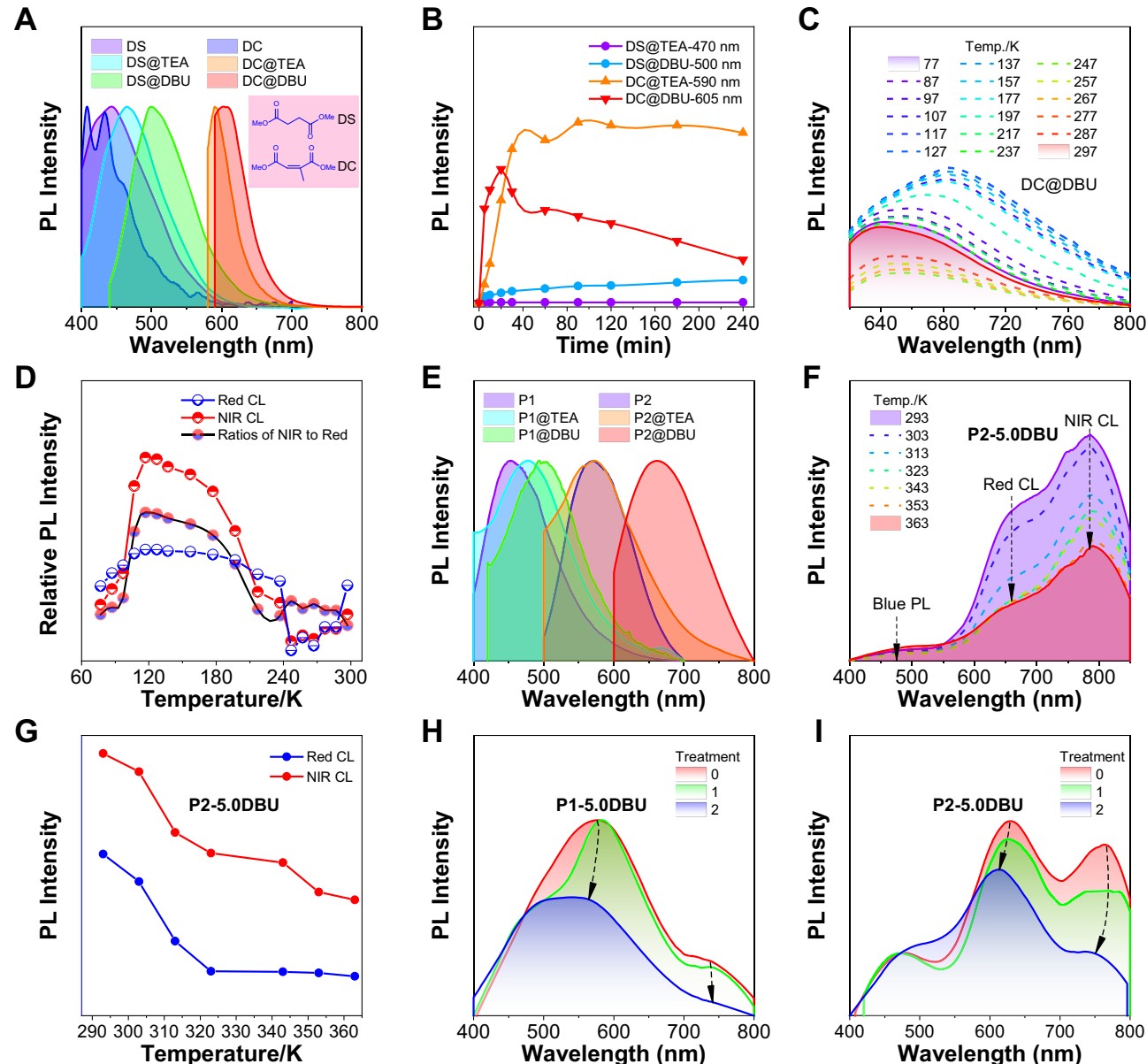

**Fig. 4 | Photophysical properties and origins of red-to-NIR CL based on amine-ester complexes. A** Normalized PL spectra of DS, **DS@TEA**, **DS@DBU**, DC, **DC@TEA**, **DC@DBU** in bulk at optimal $\lambda_{ex}$ = 360, 380, 440, 360, 560, and 560 nm, respectively. The molar ratios of these ester-containing structures to amines are 640:1. @ represents the mixing process. **B** Plots of PL intensities versus treatment time for **DS@TEA**, **DS@DBU**, **DC@TEA**, and **DC@DBU**, respectively. The molar ratios of these ester-containing structures to amines are 640:1. **C** Temperature-dependent PL spectra and **D** plots of relative PL intensities of **DC@DBU** versus the temperature in bulk. The molar ratio of DC to DBU is 20:1. $\lambda_{ex}$ = 600 nm. **E** Normalized PL spectra of **P1**, **P1@TEA**, **P1@DBU**, **P2**, **P2@TEA**, **P2@DBU** in solid at optimal $\lambda_{ex}$ = 400, 380, 400, 480, 480 and 580 nm, respectively. These mixtures were obtained by soaking polyesters in amine. **F** Temperature-dependent PL spectra and **G** Plots of PL intensities versus temperature of **P2-5.0DBU** in dimethyl sulfoxide solution of $10^{-1}$ M. **H** PL spectra of **P1-5.0DBU** and **I** PL spectra of **P2-5.0DBU** in solid precipitated in ethanol. Treatments 0, 1, and 2 represent once-precipitated treatment without hydrochloric acid, once- and twice-precipitated treatment under hydrochloric acid of 37 wt% (5 drops of hydrochloric acid per precipitation).

dissociated, resulting in the dominant red CL of short-range complexes. These results indicate that NIR CL was more sensitive to temperature compared with red CL, which should originate from long-range amine-ester complexes.

Afterward, red-to-NIR CL from amine-ester complexes dependent on chain structures was further investigated. **P2@TEA** and **P2@DBU** with folding polymer chains displayed redder CL than these of **P1@TEA** and **P1@DBU** with helical structures, both in solid and solution states (Fig. 4E and Supplementary Figs. 219-222). Meanwhile, compared with PL spectra of **DS@TEA**, **DS@DBU**, **DC@TEA,** and **DC@DBU** in Fig. 4A, the PL spectra of **P1@TEA**, **P1@DBU**, **P2@TEA** and **P2@DBU** showed redder CL, especially **P2@DBU** displayed

distinct NIR CL band at 680–800 nm, suggesting that polymers can form stable amine-ester complexes relative to small molecules. Moreover, temperature-dependent PL spectra of DBU-initiated **P2-5.0DBU** with three PL peaks (blue light at 470 nm, red CL at 660 nm, and NIR CL at 780 nm) were measured from 293 K to 363 K (Fig. 4F, G). The intensity of blue PL at 470 nm shows almost no change, while the red CL and NIR CL gradually decrease, suggesting that the red-to-NIR CL of polyesters is based on TSI from amine-ester complexes. It is noteworthy that the intensity of NIR CL keeps decreasing from 293 K to 363 K, but the red CL was stabilized above 320 K, demonstrating the higher sensitivity of NIR CL corresponding to long-range amine-ester complexes. Furthermore, red-to-NIR CL was weakened and blue-

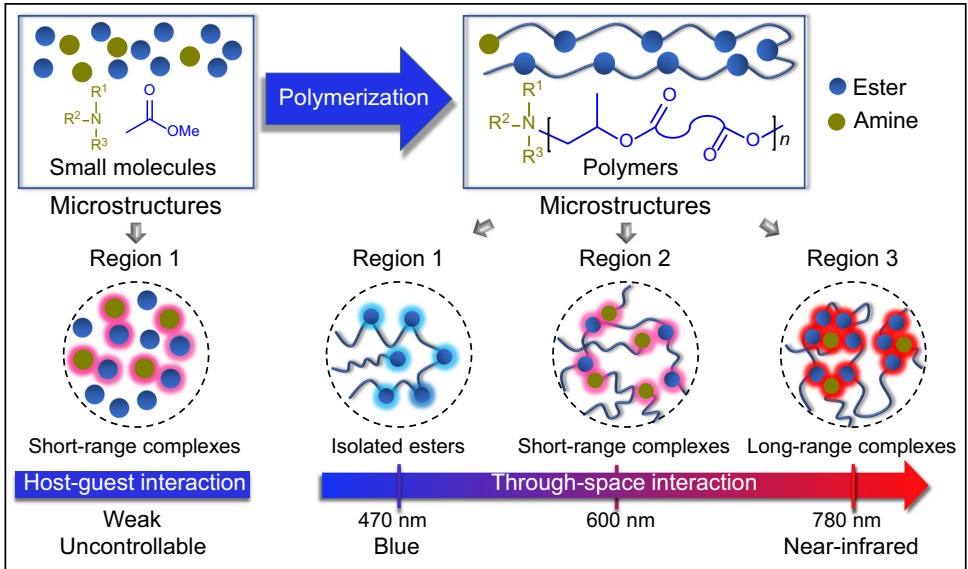

**Fig. 5 | Proposed mechanism of blue-to-NIR CL in amine-initiated polyesters.** Amine-initiated polyesters with diverse microstructures make it easier to form various amine-ester complexes to induce red-to-NIR CL. Blue and dark-green balls represent ester units and amines, respectively. Blue, pink and red light around aggregates of balls represent the emission color of isolated ester groups, short-range and long-range complexes, respectively.

shifted after eluting terminal amines of **P1-5.0DBU**, **P2-5.0DBU**, **P1-5.0TEA**, and **P2-5.0TEA** with hydrochloric acid to break complexes (Fig. 4H, I and Supplementary Figs. 223–226), which also prove the important role of amine-ester complexes in the red-to-NIR CL.

## Discussion

We prepared a series of nonconjugated polyesters with tunable, blue-to-NIR emission inspired by the appearing color phenomenon during amine-initiated polymerization. Photophysical characterization and theoretical calculation reveal that red-to-NIR CL (600–780 nm) corresponds to amine-ester complexes. Meanwhile, the underlying emission mechanism of these amine-ester complexes is intra/interchain charge transfer. The emission colors can be manipulated by molar ratios and types of amine catalysts, chain microstructures of polyesters and concentration. More importantly, the formation of amine-ester complexes was disclosed by dynamic NMR and photophysical characterization. Meanwhile, NIR CL originating from long-range amine-ester complexes was further confirmed. In general, amine-initiated polyesters with diverse microstructures make it easier to form various amine-ester complexes to induce red-to-NIR CL, suggesting the unique microstructures of polymer chains relative to small molecules in manipulating the photophysical properties (Fig. 5)[40,41].

In conclusion, we not only realize the red-to-NIR CL from non-conjugated polyesters but also provide a sustainable platform to achieve high-efficiency, controllable blue-to-NIR CLgens with deeper sight into CL.

## Methods
### Materials
All manipulations involving air- and/or water-sensitive compounds were carried out with the standard Schlenk and vacuum line techniques under an argon atmosphere or in a nitrogen-filled glovebox. Succinic anhydride (SA) and citraconic anhydride (CA) were purchased from Macklin. Propylene epoxide (PO) was purchased from Aldrich. Chromatographic purity solvents, including toluene, dichloromethane, tetrahydrofuran (THF), ethanol, and flaky sodium hydroxide (CaH2), were bought from SINOPHARM. Triethyl borane (TEB) in tetrahydrofuran solution (1.0 mol/L), Dimethyl succinate (DS), and dimethyl citraconate (DC) were purchased from the Tokyo Chemical

Industry (TCI). Triethylamine (TEA) was purchased from Aldrich, and 1,8-diazabicyclo[5.4.0]undec-7-ene (DBU), 7-methyl-1,5,7-triazabicyclo[4.4.0]dec-5-ene (mTBD) and 1,5,7-triazabicyclo[4.4.0]dec-5-ene (TBD) were purchased from TCI. PO, TEA, DBU, DS, DC, and toluene were refluxed over CaH2 for 24 h and vacuum-distilled prior to use. In particular, chloroform-d for dynamic NMR characterization of amine-ester complexes was refluxed over CaH2 for 24 h and distilled prior to use. Both the anhydrides (SA and CA) were purified by vacuum sublimation three times. The purified anhydrides were collected under an inert atmosphere and stored in the glovebox for use. Other organic reagents were used without purification.

### Synthesis
**P1** and **P2** were obtained by the copolymerization of PO with SA and CA, respectively, which was catalyzed by heterogeneous inorganic Zn–Co$^{III}$ DMCC without any organic amines. **P1-aTEA/DBU/mTBD/TBD** and **P2-aTEA/DBU/mTBD/TBD** catalyzed by organic Lewis acid-base pairs of triethylborane (TEB) with a different amine such as triethylamine (TEA), 1,8-diazabicyclo[5.4.0]undec-7-ene (DBU), 7-methyl-1,5,7-triazabicyclo[4.4.0]dec-5-ene (mTBD) and 1,5,7-triazabicyclo[4.4.0]dec-5-ene (TBD), were obtained by using different molar ratios (a%) of amines from 0.5% to 5.0% (seeing the Supporting Information for the detailed synthetic procedures)

## Data availability
All the data supporting the findings in this work are available within the manuscript and Supplementary Information file and available from the corresponding authors upon request.

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

## Acknowledgements

We gratefully acknowledge the financial support of the National Science Foundation of China (No. 51973190, received by X.Z.), Zhejiang Provincial Department of Science and Technology (No. 2020R52006, received by X.Z.), National Science Foundation of China (No. 22205197, received by H.Z.) and the support of the Youth Talent Excellence Program of ZJU-Hangzhou Global Scientific and Technological Innovation Center (to H.Z.).

## Author contributions

B.C. conceived the idea. B.C., H.Z., and X.Z. designed the experiments. B.C. and X.L. performed the synthesis. B.C., X.L., and B.L. did the photophysical measurements and analyzed the data. Z.X. conducted theoretical calculations. B.C., Z.Z., C.Z., J.S., Q.Y., H.Z., X.Z., and B.Z.T. took part in the discussion and gave important suggestions. B.C., H.Z., and X.Z. co-wrote the paper. All authors approved the final version of the paper.

## Competing interests

The authors declare no competing interests.
