## [Peer Review File · Nature Communications]

Enabling nonconjugated polyesters emit full-spectrum fluorescence from blue to near-infraredREVIEWER COMMENTS

Reviewer #1 (Remarks to the Author):

In this manuscript, zhang and coworkers reported an engineering strategy to regulate blue-to-NIR light by constructing amine-ester complexes in polyesters, inspired by the appearance color phenomenon during organocatalyzed polymerization. Through a series of molecular, photophysical characterization and theoretical calculations, the authors revealed that amines acted as initiators attached to polymer chain ends and formed stable amine-ester complexes, which induce red-to-NIR CL at 600–780 nm via intra-/inter-chain charge transfer. Meanwhile, the emission colors can be manipulated by content and types of amines, chain microstructures of polyesters and their concentration. Further, the formation of amine-ester complexes was disclosed by dynamic NMR and photophysical characterization. In addition, the NIR light originating from long-range amine-ester complexes was further confirmed, indicating polyesters with diverse microstructures are easier to form various amine-ester complexes. The phenomenon in this work is interesting and the author gives a reliable mechanism picture for the NIR-emitter construction by complexation. The work is innovative and well-justified. Therefore, this manuscript is recommended for publication subject to the following minor comments.

- 1, Characterization of complex dynamic NMR spectra should be described in detail in Supporting Information.
- 2, In the first part of the results “The polyesters P1 and P2 were obtained by the copolymerization of epoxides and cyclic anhydrides, which were catalyzed by heterogeneous inorganic zinc-cobalt (III) double metal cyanide complex (Zn-CoIII DMCC) as the control group”. Since the Zn-catalyzed polyesters can be used as the control group, please state in the text that Zn-OH does not interfere with luminescence or cite previous literature as support.
- 3, In the sentence: “(ii) In the range of c_1 to c_2 , the complexes began to form and the red-CL was induced and enhanced with further increasing of c ”, authors use the word “red-CL”, but use “red CL” in other places. Please use it consistently.
- 4, In supplementary Figure S19-21, do the 2' and 4' represent the ^1H chemical shifts of ether linkages from the continuous ring-opening reaction of propylene oxide? Please mark clearly in these Figures.
- 5, Whether small content of ether linkages interfere with luminescence?

Reviewer #2 (Remarks to the Author):

The authors report in the manuscript that non-conjugated polyester can show non-conventional luminescence from blue to NIR region in the presence of amines. From the series of mechanistic studies, the authors claim that amin-polyester complexes through both intra/intermolecular interactions should play a critical role in emission. It is the first example to offer NIR emission without non-conjugated organic materials. Therefore, scientific novelty is high enough for publication in this journal. However, I have several unclear issues in the current version of the manuscript. If these issues are appropriately

supported, I think that this manuscript is suitable for publication:

-It has been confirmed that the light emission disappears when an acid is added during the formation of an amine-ester complex. Is it possible to induce annihilation by temperature rising above room temperature? These data can also represent the thermal stability of luminescent properties of your materials.

-According to the calculation, although the authors show the optimized structures involving amine-ester complexes, there should be some with different orientations. Did you investigate energy levels in semi-stable structures?

-DFT and TD-DFT calculations were performed without CAM, despite the fact that charge transfer is the important process. The authors should compare the calculation results with CAM.

-Do you have information on the amine effect with diisopropylethylamine instead of TEA.

-Please add further explanation for using bicycloamines such as DBU. What happens if cycloamine is used?

Reviewer #3 (Remarks to the Author):

In this work, Zhang, et.al reported a series of aliphatic polyesters that were formed from amine initiation, and the authors proposed a simple and useful method of using the end amine group complexed with the ester groups, which transformed polyesters into blue-to-NIR luminophores. The extensive experiments supported the proposal of the complexation between the end amine group and the ester groups in chain. Such complexation is responsible to the red-to-NIR CL at 600–780 nm, through intra- / inter- chain charge transfer. More importantly, the authors further confirmed the NIR CL in these polyesters originating from long-range amine-ester complexes, and they revealed that diverse microstructures of these polymers induced the formation of various amine-esters complexes, relative to small molecules in manipulating the photophysical properties.

Currently, it is still a big problem and rather difficult in understanding the heteroatom-rich nonconjugated polymers emitting visible lights, let alone the NIR light. This reviewer thought that the nonconjugated polymers are very promising next-generation luminophores owing to the advantages of mass production, easy processing and biocompatibility. This work, of course, is an important and nice contribution to the photophysics and materials as it enables nonconjugated polyesters to emit NIR light for the first time, as well as the insight into the photophysical mechanism. In methodology, this manuscript provides a new strategy to design nonconjugated luminophores with the emission of NIR light. As a result, this reviewer is particularly happy to recommend the publication of this work. Some issues should be revised or improved:

1. In Supplementary Figures 52 and 53, the P2-1.0TEA~ P2-1.0TEA should be corrected to P1-1.0TEA~ P1-

1. OTEA, and P2-2.0mTBD~ P2-2.0TBBD should be corrected to P1-2.0mTBD~ P1-2.0TBBD. Please double check the supporting information of this work.

2. Does the red-to-NIR light in amine-initiated polyesters come from the complexation of the trace free amine molecules and polyesters?

3. From the perspective of polymerization, augmenting the organic base content and molecular weight without the production of ether linkages seems unjustifiable. The NMR results have hydrogen chemical shifts of ether linkages under high amine contents, but Table S1 and S2 show ~99% ester linkages without any ether linkages in polymers. Please correct the contents of ester linkages according to NMR results.

Replies to Reviewer(s)' Comments

Reviewer 1: *In this manuscript, zhang and coworkers reported an engineering strategy to regulate blue-to-NIR light by constructing amine-ester complexes in polyesters, inspired by the appearance color phenomenon during organocatalyzed polymerization. Through a series of molecular, photophysical characterization and theoretical calculations, the authors revealed that amines acted as initiators attached to polymer chain ends and formed stable amine-ester complexes, which induce red-to-NIR CL at 600–780 nm via intra-/inter-chain charge transfer. Meanwhile, the emission colors can be manipulated by content and types of amines, chain microstructures of polyesters and their concentration. Futher, the formation of amine-ester complexes was disclosed by dynamic NMR and photophysical characterization. In addition, the NIR light originating from long-range amine-ester complexes was further confirmed, indicating polyesters with diverse microstructures are easier to form various amine-ester complexes. The phenomenon in this work is interesting and the author gives a reliable mechanism picture for the NIR-emitter construction by complexation. The work is innovative and well-justified. Therefore, this manuscript is recommended for publication subject to the following minor comments.*

Reply: Thank you for your encouraging comments.

1. *Characterization of complex dynamic NMR spectra should be described in detail in Supporting Information.*

Reply: Thanks for your suggestions. Characterization of complex dynamic NMR spectra have been described in detail in Supporting Information.

“Dynamic NMR Spectra of Complexation

P1@DBU: **P2** (17 mg, 0.10 mmol) and **DBU** (3 μ L, 0.02mmol) were dissolved by 0.5 mL of anhydrous chloroform-d in clean glass bottles, then the solution in bottle was transferred into one NMR tube for ^1H - and ^{13}C -NMR characterization on a Bruker 400 MHz instrument by using tetramethylsilane (TMS) as an internal reference. The

measurement times were at 1 hour, 1 day, 2 days, 3 days, 4 days and 5 days after dissolution, respectively. $\Delta\delta_{\text{H}}$ and $\Delta\delta_{\text{C}}$ represent the absolute value of chemical shift changes for hydrogen and carbon atoms on DBU.

DS@DBU: P1 (13 μL , 0.1 mmol) and **DBU** (3 μL , 0.02mmol) were dissolved by 0.5 mL of anhydrous chloroform-d in clean glass bottles, then the solution in bottle was transferred into one NMR tube for ^1H - and ^{13}C -NMR characterization on a Bruker 400 MHz instrument by using tetramethylsilane (TMS) as an internal reference. The measurement times were at 1 hour, 1 day, 2 days, 3 days and 4 days after dissolution, respectively. $\Delta\delta_{\text{H}}$ and $\Delta\delta_{\text{C}}$ represent the absolute value of chemical shift changes for hydrogen and carbon atoms on DBU.

DC@DBU: P1 (14 μL , 0.1 mmol) and **DBU** (3 μL , 0.02mmol) were dissolved by 0.5 mL of anhydrous chloroform-d in clean glass bottles, then the solution in bottle was transferred into one NMR tube for ^1H - and ^{13}C -NMR characterization on a Bruker 400 MHz instrument by using tetramethylsilane (TMS) as an internal reference. The measurement times were at 1 hour, 1 day, 2 days, 3 days and 4 days after dissolution, respectively. $\Delta\delta_{\text{H}}$ and $\Delta\delta_{\text{C}}$ represent the absolute value of chemical shift changes for hydrogen and carbon atoms on DBU.

CA@DBU: The different mixtures with various molar ratios of **CA** to **DBU** were dissolved by 0.5 mL of anhydrous chloroform-d in clean glass bottles, then the solution in bottle was transferred into NMR tubes for ^1H - and ^{13}C -NMR characterization on a Bruker 400 MHz instrument by using tetramethylsilane (TMS) as an internal reference. The molar ratios of CA to DBU were 5:0, 1:1, 1:5 and 0:5, respectively. $\Delta\delta_{\text{H}}$ and $\Delta\delta_{\text{C}}$ represent the absolute value of chemical shift changes for hydrogen and carbon atoms on DBU.

SA@DBU: The different mixtures with various molar ratios of **CA** to **DBU** were dissolved by 0.5 mL of anhydrous chloroform-d in clean glass bottles, then the solution in bottle was transferred into NMR tubes for ^1H - and ^{13}C -NMR characterization on a Bruker 400 MHz instrument by using tetramethylsilane (TMS) as an internal reference. The molar ratios of CA to DBU were 5:0, 5:1, 4:1, 3:1, 2:1, 1:1, 1:2, 1:3, 1:4, 1:5 and 0:5, respectively. $\Delta\delta_{\text{H}}$ and $\Delta\delta_{\text{C}}$ represent the absolute value of chemical shift changes

for hydrogen and carbon atoms on DBU.

P1@TEA: P1 (15.8 mg, 0.10 mmol) and TEA (2.8 μ L, 0.02mmol) were dissolved by 0.5 mL of anhydrous chloroform-d in clean glass bottles, then the solution in bottle was transferred into one NMR tube for ^1H - and ^{13}C -NMR characterization on a Bruker 400 MHz instrument by using tetramethylsilane (TMS) as an internal reference. The measurement times were at 1 hour, 1 day, 2 days, 3 days, 4 days and 5 days after dissolution, respectively.

P2@TEA: P1 (17 mg, 0.10 mmol) and TEA (2.8 μ L, 0.02mmol) were dissolved by 0.5 mL of anhydrous chloroform-d in clean glass bottles, then the solution in bottles bottle was transferred into one NMR tube for ^1H - and ^{13}C -NMR characterization on a Bruker 400 MHz instrument by using tetramethylsilane (TMS) as an internal reference. The measurement times were at 1 hour, 1 day, 2 days, 3 days, 4 days and 5 days after dissolution, respectively.

DS@TEA: P1 (13 μ L, 0.10 mmol) and TEA (2.8 μ L, 0.02mmol) were dissolved by 0.5 mL of anhydrous chloroform-d in clean glass bottles, then the solution in bottles bottle was transferred into one NMR tube for ^1H - and ^{13}C -NMR characterization on a Bruker 400 MHz instrument by using tetramethylsilane (TMS) as an internal reference. The measurement times were at 1 hour, 1 day, 2 days, 3 days and 4 days after dissolution, respectively.

DC@TEA: P1 (14 μ L, 0.10 mmol) and TEA (2.8 μ L, 0.02mmol) were dissolved by 0.5 mL of anhydrous chloroform-d in clean glass bottles, then the solution in bottle was transferred into one NMR tube for ^1H - and ^{13}C -NMR characterization on a Bruker 400 MHz instrument by using tetramethylsilane (TMS) as an internal reference. The measurement times were at 1 hour, 1 day, 2 days, 3 days and 4 days after dissolution, respectively.”

2. *In the first part of the results “The polyesters P1 and P2 were obtained by the copolymerization of epoxides and cyclic anhydrides, which were catalyzed by heterogeneous inorganic zinc-cobalt (III) double metal cyanide complex (Zn–CoIII DMCC) as the control group”. Since the Zn-catalyzed polyesters can be used as the control group, please state in the text that Zn-OH does not interfere with luminescence*

or cite previous literature as support.

Reply: Thanks for your suggestions. Our previous work has completely ruled out the effect of trace impurities such as catalysts on PL for such similar polyesters by conducting control experiments. So, we add the statement in the main text that “*which were catalyzed by heterogeneous inorganic zinc-cobalt (III) double metal cyanide complex (Zn–CoIII DMCC)³⁶ as the control group with no interference of Zn-OH on luminescence^{33,34}*”. Also, the relative literature “(33) *Chu, B. et al. Aliphatic Polyesters with White-Light Clusteroluminescence. J. Am. Chem. Soc. 144, 15286-15294 (2022).* (34) *Chu, B. et al. Altering Chain Flexibility of Aliphatic Polyesters for Yellow-Green Clusteroluminescence in 38 % Quantum Yield. Angew. Chem. Int. Ed. 61, e202114117 (2022)*” have been cited.

3. *In the sentence: “ii) In the range of c1 to c2, the complexes began to form and the red-CL was induced and enhanced with further increasing of c”, authors use the word “red-CL”, but use “red CL” in other places. Please use it consistently.*

Reply: Thanks for your suggestions. We have consistently used the word “red CL” in the revised manuscript.

4. *In supplementary Figure S19-21, do the 2’ and 4’ represent the ¹H chemical shifts of ether linkages from the continuous ring-opening reaction of propylene oxide? Please mark clearly in these Figures.*

Reply: Thanks for your professional suggestions. The ¹H chemical shifts 2’ and 4’ represent the small content of linkages and we have marked clearly in Figures S19 and S21.

Supplementary Figure 19. ¹H NMR spectra of polyester **P1-3.0TEA** (400 MHz, CDCl₃).

Supplementary Figure 21. ¹H NMR spectra of polyester **P1-4.0TEA** (400 MHz, CDCl₃).

5. Whether small content of ether linkages interfere with luminescence?

Reply: Thanks for your comment.

According to our previous work, “*Chu, B. et al. Altering Chain Flexibility of Aliphatic Polyesters for Yellow-Green Clusteroluminescence in 38 % Quantum Yield. Angew. Chem. Int. Ed. 61, e202114117 (2022)*”, the randomly distributed ether units in similar polyesters by conducting control experiments only influence QYs by changing flexibility of polyester backbones, rather than altering emission wavelengths.

In this work, we also investigated the photophysical properties of ether units in pure polyether (PPO-2.0TEA) from homopolymerization of PO catalyzed by TEB/TEA in Figure R1. PPO-2.0TEA in solid only showed blue light around 430 nm without any red-to-NIR CL, which rules out the possibility that ether units participate in the complexation.

Figure R1. **A** ¹H and **B** ¹³C spectra of polyester **PPO-2.0TEA** from homopolymerization of PO catalyzed by TEB/TEA (400MHz, CDCl₃). **C** MALDI-TOF mass spectrum of P1-2.0DBU. **D** PL spectrum of **PPO-2.0TEA** in solid.

Reviewer 2: *The authors report in the manuscript that non-conjugated polyester can show non-conventional luminescence from blue to NIR region in the presence of amines. From the series of mechanistic studies, the authors claim that amine-polyester complexes through both intra/intermolecular interactions should play a critical role in emission. It is the first example to offer NIR emission without non-conjugated organic materials. Therefore, scientific novelty is high enough for publication in this journal. However, I have several unclear issues in the current version of the manuscript. If these issues are appropriately supported, I think that this manuscript is suitable for publication:*

Reply: Thank you for your encouraging comments.

1. *It has been confirmed that the light emission disappears when an acid is added during the formation of an amine-ester complex. Is it possible to induce annihilation by temperature rising above room temperature? These data can also represent the thermal stability of luminescent properties of your materials.*

Reply: Thanks for your professional suggestions. In Figures 4F and 4G, we have set P2-5.0DBU as an example to study the thermal stability of red-to-NIR CL of our materials. Temperature-dependent PL spectra and Plots of PL intensities versus temperature of **P2-5.0DBU** indicate that red-to-NIR CL only decreased with increasing temperature from 293 K to 323 K, and remains unchanged under 323K~363K. Hence, the annihilation effect via increasing temperature was difficult to achieve. It is worth noting that the red-to-NIR CL still exists without any wavelength shift even at a higher temperature of 363 K (90 °C), suggesting the good thermal stability of the luminescent properties of our materials.

Figure 4. **A** Normalized PL spectra of DS, DS@TEA, DS@DBU, DC, DC@TEA, DC@DBU in bulk at optimal $\lambda_{ex} = 360, 380, 440, 360, 560$ and 560 nm, respectively. The molar ratios of these ester-containing structures to amines are 640:1. @ represents the mixing process. **B** Plots of PL intensities versus treatment time for DS@TEA, DS@DBU, DC@TEA, DC@DBU, respectively. The molar ratios of these ester-containing structures to amines are 640:1. **C** Temperature-dependent PL spectra and **D** plots of relative PL intensities of DC@DBU versus the temperature in bulk. The molar ratio of DC to DBU is 20:1. $\lambda_{ex} = 600$ nm. **E** Normalized PL spectra of P1, P1@TEA, P1@DBU, P2, P2@TEA, P2@DBU in solid at optimal $\lambda_{ex} = 400, 380, 400, 480, 480$ and 580 nm, respectively. These mixtures were obtained by soaking polyesters in amine. **F** Temperature-dependent PL spectra and **G** Plots of PL intensities versus temperature of P2-5.0DBU in dimethyl sulfoxide solution of 10^{-1} M. **H** PL spectra of P1-5.0DBU and **I** PL spectra of P2-5.0DBU in solid precipitated in ethanol. Treatment 0, 1, 2 represent once-precipitated treatment without hydrochloric acid, once- and twice-precipitated treatment under hydrochloric acid of 37 wt% (5 drops of hydrochloric acid per precipitation).

2. According to the calculation, although the authors show the optimized structures involving amine-ester complexes, there should be some with different orientations. Did you investigate energy levels in semi-stable structures?

Reply: Thanks for your suggestions. We set **P1@DBU** as an example and provide two semi-stable structures in different orientations in Figure R2. By comparing the semi-stable structures (Figure R2) and optimized excited-state geometry (Figure R3), although their orientations are different, the charge transfer process has always existed, suggesting the strong complex between amine and ester. However, the optimized excited-state geometry has a narrower energy gap than the semi-stable structures, indicating the important role of amine-ester complexes in regulating the luminescent properties.

Figure R2. Frontier molecular orbitals of excited-state geometries of semi-stable structures of **P1@DBU** calculated by TD-DFT method at CAM-B3LYP-D3/6-31G(d,p) level, Gaussian 09 program.

Figure R3. Frontier molecular orbitals of optimized excited-state geometry of **P1@DBU** calculated by TD-DFT method at CAM-B3LYP-D3/6-31G(d,p) level, Gaussian 09 program.

3. *DFT and TD-DFT calculations were performed without CAM, despite the fact that charge transfer is the important process. The authors should compare the calculation results with CAM.*

Reply: Thanks for your suggestions. TD-DFT calculations were performed with CAM were carried out to optimize the excited-state geometries of **P1@DBU**. In Figure R3, based on calculation with CAM, the HOMO and LUMO were located on nitrogen atoms of DBU and esters of polyesters, respectively, which is similar to the results without CAM in Figure 3G.

4. *Do you have information on the amine effect with diisopropylethylamine instead of TEA.*

Reply: Thanks for your suggestions. We set **P1-2.0DIPEA** as the example to investigate the amine effect. The **P1-2.0DIPEA** is copolymerized of PO and anhydrides initiated by diisopropylethylamine (DIPEA). In Figures R4 and R5, the chemical structure of **P1-2.0DIPEA** is confirmed. We then investigate its PL properties as shown in Figure R6-R8. In Figure R6, with the concentration increasing from 10^{-5} to 10^{-1} M, a new absorption band at 300~600 nm appeared and its intensity was gradually enhanced, suggesting that complexation between DIPEA and esters induced a new

strong TSIs. Meanwhile, in Figure R7, the excitation spectra of **P1-2.0DIPEA** in DCM solution and solid state matched well with the absorption band at 300~600 nm, indicating that the dominant excited centers originate from complexes of DIPEA and esters. Also, the PL spectra displayed the long-wavelength CL at 500~600 nm from complexes of DIPEA and esters (Figure R8). These results indicate that DIPEA has the same amine effect as TEA on the formation of amine-ester complexes and their CL properties.

Figure R4. ¹H NMR spectra of polyester **P1-2.0DIPEA** in CDCl₃ (400 MHz, CDCl₃).

Figure R5. ¹³C NMR spectra of polyester **P1-2.0 DIPEA** in CDCl₃ (100 MHz, CDCl₃).

Figure R6. Concentration-dependent UV-Vis absorption spectra of **P1-2.0 DIPEA** in DCM solution. Concentration: from 10⁻⁵ to 10⁻¹ M.

Figure R7. Excitation spectra under different emission wavelengths of **P1-2.0 DIPEA** in (A) DCM solution of 10^{-1} M and (B) solid state.

Figure R8. PL spectra under different excitation wavelengths of **P1-2.0 DIPEA** in (A) DCM solution of 10^{-1} M and (B) solid state.

5. Please add further explanation for using bicycloamines such as DBU. What happens if cycloamine is used?

Reply: Thanks for your good suggestions.

1. In this work, we have confirmed that nitrogen on amines played an important role in amine-ester complexes by comparing four different organic amines (TEA, DBU, mTBD and TBD), which all exhibited the same red-NIR CL peaks (Fig. 3A and Fig. S160 –162). So in the case of DBU of bicycloamines, the two nitrogen atoms on it are responsible for the amine-ester complexes. To figure out which of the two nitrogen

atoms has dominant electronic interaction with esters, we carried out a series of time-dependent ^1H - and ^{13}C -NMR spectra of **P1@DBU**, **P2@DBU**, **DS@DBU**, **DC@DBU**, **CA@DBU** and **SA@DBU**. In Figure 3C-3F and Figure S163-176, The maximum $\Delta\delta_{\text{H}}$ and $\Delta\delta_{\text{C}}$ were located at positions 4 or 5, while positions 2 and 3 had only negligible change, indicating that the nitrogen atom closed to positions 4 and 5 on DBU forms complexes with ester units. Especially, Ratio-dependent NMR spectra of **SA@DBU** disclosed the molar complexation ratio of DBU to SA is 2:1 (Figure S177–197), which suggests that one nitrogen of DBU complexes with one carbonyl or ester.

To further confirm how nitrogen on DBU interacts with ester to induce red-to-NIR CL, we calculated the electronic natures of polyesters after introducing DBU. Theoretical calculations were carried out to optimize the excited-state geometries of **P1@DBU**, **P2@DBU**, **P1-DBU** and **P2-DBU**, and these results suggested that the electron transitions occur between DBU and ester moieties (Figure 3G-3H, Figure S206-207). Meanwhile, the hole (red color) and electron (blue color) of optimized excited-state geometries of **P1@DBU**, **P2@DBU**, **P1-DBU** and **P2-DBU** were further simulated (Fig. 3H and Fig. S207). The low-lying excited state exhibits a through-space charge transfer (TSCT) effect among the DBU and esters.

Overall, after introducing DBU, the nitrogen closed to positions 4 and 5 on DBU played an important role in amine-ester complexes by intra/interchain charge transfer or through-space charge transfer (TSCT) effect.

2. we also investigated amine effect of cycloamine such as typical 1-methylpiperidine (MP), and obtained **P1-2.0MP** copolymerized of PO and anhydrides initiated by MP. In Figures R9 and R10, the chemical structure of **P1-2.0MP** is confirmed. We then investigate its PL properties as shown in Figure R11-R13. In Figure R11, with the concentration increasing from 10^{-5} to 10^{-1} M, a new absorption band at 300~600 nm appeared and its intensity was gradually enhanced, suggesting that complexation between MP and esters induced a new strong TSIs. Meanwhile, in Figure R12, the excitation spectra of **P1-2.0MP** in DCM solution and solid state matched well with the absorption band at 300~600 nm, indicating that the dominant excited centers originate from complexes of MP and esters. Also, the PL spectra displayed the long-

wavelength CL at 500~600 nm from complexes of MP and esters (Figure R13). These results indicate that cycloamine MP has the same amine effect as dicycloamines.

Figure 3. **A** Normalized PL spectra of **P1-2.0TEA**, **P1-2.0DBU**, **P1-2.0mTBD** and **P1-2.0TBD** in solid at $\lambda_{\text{ex}} = 360$ nm. Inset: photographs taken under 365 nm UV light of **P1-2.0TEA**, **P1-2.0DBU**, **P1-2.0mTBD** and **P1-2.0TBD** from left to right. **B** PL spectra of **P1-5.0DBU** in the solid state recorded at different excitation wavelengths. **C** Time-dependent $^1\text{H-NMR}$ and **D** $^{13}\text{C-NMR}$ spectra of **P1@DBU** obtained from mixing of **P1** and DBU. The molar ratio of structural units of **P1** to DBU is 5:1. @ represent the symbol of mixing process. The h and d represent hours and days, respectively. **E** and **F** Plots of absolute value of chemical shift changes ($\Delta\delta_{\text{H}}$ for hydrogen atoms, $\Delta\delta_{\text{C}}$ for carbon atoms) versus atom positions on DBU for **P1@DBU** and **DS@DBU**, respectively. **DS** is the model compound of **P1**. The molar ratio of **DS** to DBU is 5:1. **G** Frontier molecular orbitals and **H** Hole (red color) and electron (blue color) of optimized excited-state geometries of **P1@DBU** and **P1-DBU** calculated by TD-DFT method at B3LYP-D3/6-31G(d,p) level, Gaussian 09 program. HOMO: the highest

occupied molecular orbital, LUMO: the lowest unoccupied molecular orbital.

Figure R9. ^1H NMR spectra of polyester **P1-2.0MP** in CDCl_3 (400 MHz, CDCl_3).

Figure R10. ^{13}C NMR spectra of polyester **P1-2.0MP** in CDCl_3 (100 MHz, CDCl_3).

Figure R11. Concentration-dependent UV-Vis absorption spectra of **P1-2.0MP** in DCM solution. Concentration: from 10^{-5} to 10^{-1} M.

Figure R12. Excitation spectra under different emission wavelengths of **P1-2.0MP** in (A) DCM solution of 10^{-1} M and (B) solid state.

Figure R13. PL spectra under different excitation wavelengths of **P1-2.0 MP** in (A) DCM solution of 10^{-1} M and (B) solid state.

Reviewer 3: *In this work, Zhang, et.al reported a series of aliphatic polyesters that were formed from amine initiation, and the authors proposed a simple and useful method of using the end amine group complexed with the ester groups, which transformed polyesters into blue-to-NIR luminophores. The extensive experiments supported the proposal of the complexation between the end amine group and the ester groups in chain. Such complexation is responsible to the red-to-NIR CL at 600–780 nm, through intra- / inter- chain charge transfer. More importantly, the authors further confirmed the NIR CL in these polyesters originating from long-range amine-ester complexes, and they revealed that diverse microstructures of these polymers induced the formation of various amine-esters complexes, relative to small molecules in manipulating the photophysical properties.*

Currently, it is still a big problem and rather difficult in understanding the heteroatom-rich nonconjugated polymers emitting visible lights, let alone the NIR light. This reviewer thought that the nonconjugated polymers are very promising next-generation luminophores owing to the advantages of mass production, easy processing and biocompatibility. This work, of course, is an important and nice contribution to the photophysics and materials as it enables nonconjugated polyesters to emit NIR light for the first time, as well as the insight into the photophysical mechanism. In methodology, this manuscript provides a new strategy to design nonconjugated luminophores with the emission of NIR light. As a result, this reviewer is particularly happy to recommend the publication of this work.

Reply: Thank you for your encouraging comments.

Some issues should be revised or improved:

1. In Supplementary Figures 52 and 53, the P2-1.0TEA~ P2-1.0TEA should be corrected to P1-1.0TEA~ P1-1.0TEA, and P2-2.0mTBD~ P2-2.0TBD should be corrected to P1-2.0mTBD~ P1-2.0TBD. Please double check the supporting information of this work.

Reply: Thanks for your suggestions. These mistakes in Supplementary Figures 52 and 53 have been corrected.

2. Does the red-to-NIR light in amine-initiated polyesters come from the complexation

of the trace free amine molecules and polyesters?

Reply: Thanks for your professional suggestions.

On the one hand, we confirmed the red-to-NIR CL in these polyesters originating from various amine-ester complexes from end amines and esters in amine-capped polyesters.

On the other hand, trace-free amine molecules are inevitable since the initiation efficiency of amines is not 100%, and polymers may contain trace amounts of free amines even after purification. The complexation of the trace-free amine molecules and polyesters is equivalent to the blending process of amines and polyesters. In this manuscript, we have carried out a series of blending experiments of **P1@TEA**, **P2@TEA**, **P1@DBU** and **P2@DBU** (@ represents the mixing process) in Figure 4E, Supplementary Figures S220 and 222. In Supplementary Figure S220 and Figure 4E, **P1@TEA** and **P1@DBU** only showed short-wavelength emission at 480~510 nm, which is different from the red CL at 600 nm of amine-capped **P1-aTEA** and **P1-aDBU**. Also, **P2@TEA** and **P2@DBU** only showed short-wavelength emission at 570~660 nm, which is different from the NIR CL at 740~750 nm of amine-capped **P2-aTEA** and **P2-aDBU**.

Hence, complexes of free amines and polyesters have a negligible contribution to the red-to-NIR CL, and the red-to-NIR CL originates from complexes of terminal amines and esters in amine-initiated polyesters with diverse microstructures.

Figure 4. **A** Normalized PL spectra of DS, DS@TEA, DS@DBU, DC, DC@TEA, DC@DBU in bulk at optimal $\lambda_{ex} = 360, 380, 440, 360, 560$ and 560 nm, respectively. The molar ratios of these ester-containing structures to amines are 640:1. @ represents the mixing process. **B** Plots of PL intensities versus treatment time for DS@TEA, DS@DBU, DC@TEA, DC@DBU, respectively. The molar ratios of these ester-containing structures to amines are 640:1. **C** Temperature-dependent PL spectra and **D** plots of relative PL intensities of DC@DBU versus the temperature in bulk. The molar ratio of DC to DBU is 20:1. $\lambda_{ex} = 600$ nm. **E** Normalized PL spectra of P1, P1@TEA, P1@DBU, P2, P2@TEA, P2@DBU in solid at optimal $\lambda_{ex} = 400, 380, 400, 480, 480$ and 580 nm, respectively. These mixtures were obtained by soaking polyesters in amine. **F** Temperature-dependent PL spectra and **G** Plots of PL intensities versus temperature of P2-5.0DBU in dimethyl sulfoxide solution of 10^{-1} M. **H** PL spectra of P1-5.0DBU and **I** PL spectra of P2-5.0DBU in solid precipitated in ethanol. Treatment 0, 1, 2 represent once-precipitated treatment without hydrochloric acid, once- and twice-precipitated treatment under hydrochloric acid of 37 wt% (5 drops of hydrochloric acid per precipitation).

Supplementary Figure S220. PL spectra of (A) **P1**, (B) **P1@TEA** and (C) **P1@DBU** in solid with molar ratios of ester units to amines are 1/2 under different excitation wavelengths.

Supplementary Figure S222. PL spectra of (A) **P2**, (B) **P2@TEA** and (C) **P2@DBU** in solid with molar ratios of ester units to amines are 1/2 under different excitation wavelengths.

3. From the perspective of polymerization, augmenting the organic base content and molecular weight without the production of ether linkages seems unjustifiable. The NMR results have hydrogen chemical shifts of ether linkages under high amine contents, but Table S1 and S2 show ~99% ester linkages without any ether linkages in polymers. Please correct the contents of ester linkages according to NMR results.

Reply: Thanks for your professional suggestions. The more accurate contents of ester linkages of **P1-aTEA** according to NMR results have been revised in Tables S1 and S2.

Supplementary Table 1. Summary of physical properties of P1 and P1-amines

Sample	Ester ^a (%)	M_n^b (kg/mol)	PDI ^b	λ_{abs}^c (DCM)(nm)	λ_{em}^d (DCM)(nm)	λ_{em}^e (solid)(nm)	Φ^d (DCM, %)	Φ^e (solid, %)
P1	>99	10.8	1.38	253, 306	460	450	7.6	9.20
P1-0.5TEA	>99	5.2	1.38	247,310	437, 520	473, 587	4.9	10.4
P-1.0TEA	>99	6.9	1.69	250, 317, 364	441, 512	475, 584	4.0	13.0
P1-1.5TEA	97	2.9	1.90	253, 306, 376	467, 560	474, 598	4.6	12.2
P1-2.0TEA	93	6.9	1.58	252, 306, 374	485, 570	472, 595	5.2	11.6
P1-3.0TEA	85	6.7	1.51	353, 307, 387, 441	485, 575	467, 595	3.8	11.7
P1-4.0TEA	64	7.4	1.60	252, 305, 388,433	483, 582	475, 593	3.4	12.8
P1-5.0TEA	>99	18.2	1.55	253, 300, 386,438	489, 582	471, 602	4.0	15.8
P1-2.0DBU	>99	4.2	1.84		465, 595	468, 604	3.8	19.4
P1-2.0mTBD	>99	6.1	2.21		491,583	475, 580	3.9	12.8
P1-2.0TBD	>99	4.1	2.10		474, 580	481, 593	3.3	8.24

a: Ester contents calculated by ¹H NMR spectra. b: number-average molecular weight and polydispersity measured by GPC using THF as the eluent. c: UV-Vis absorption spectra in DCM (10⁻³ M). d: PL spectra and QYs in DCM (10⁻¹ M). e: PL spectra and QYs in solid. QYs were measured λ_{ex} =360 nm.

Supplementary Table 2. Summary of physical properties of P2 and P2-amines

Sample	Ester ^a (%)	M_n^b (kg/mol)	PDI ^b	λ_{abs}^c (DCM)(nm)	λ_{em}^d (DCM)(nm)	λ_{em}^e (solid)(nm)	Φ^d (DCM, %)	Φ^e (solid, %)
P2	>99	10.4	1.62	242, 300	490	477, 572	4.2	4.4
P2-0.5TEA	>99	3.9	2.91	294, 353	500, 613	496, 605, 717	9.6	5.0
P2-1.0TEA	>99	3.2	2.00	296, 347	500, 612	499, 614, 740	6.1	3.8
P2-1.5TEA	>99	2.7	2.78	296, 356	498, 611	494, 616, 738	7.2	2.1
P2-2.0TEA	>99	2.9	2.95	292, 353	501, 612, 722	493, 614, 745	6.5	1.9
P2-3.0TEA	>99	2.4	2.79	292, 332	649, 722	495, 617, 748	4.8	1.2
P2-4.0TEA	>99	1.7	2.78	290, 338	659, 725	497, 640, 759	2.9	1.3
P2-5.0TEA	>99	1.6	3.90	298, 350	660, 740	490, 640, 750	3.7	1.4
P2-2.0DBU	>99	4.8	2.40	290, 347	668, 730	487, 656, 747	3.7	1.0
P2-2.0mTBD	>99	8.4	1.53	294, 349	654, 730	484, 636, 727	2.9	1.2
P2-2.0TBD	>99	6.9	1.74	295, 334	648, 738	491, 650, 769	4.6	0.6

a: Ester contents calculated by ¹H NMR spectra. b: number-average molecular weight and polydispersity measured by GPC using THF as the eluent. c: UV-Vis absorption spectra in DCM (10⁻³ M). d: PL spectra and QYs in DCM (10⁻¹ M). e: PL spectra and QYs in solid. QYs were measured at $\lambda_{\text{ex}}=500$ nm.

REVIEWERS' COMMENTS

Reviewer #1 (Remarks to the Author):

The authors have addressed my concerns and I think that this manuscript is suitable for publication in Nature Communications.

Reviewer #2 (Remarks to the Author):

I think that the authors appropriately made an answer for my queries. Thus, the manuscript is now suitable for publication as is.

Reviewer #3 (Remarks to the Author):

The authors have addressed all the questions. The reviewer do not have any criticisms.

Replies to Reviewer(s)' Comments

Reviewer 1: *The authors have addressed my concerns and I think that this manuscript is suitable for publication in Nature Communications..*

Reply: Thank you for your encouraging comments.

Reviewer 2: *I think that the authors appropriately made an answer for my queries. Thus, the manuscript is now suitable for publication as is.*

Reply: Thank you for your encouraging comments.

Reviewer 3: *The authors have addressed all the questions. The reviewer do not have any criticisms.*

Reply: Thank you for your encouraging comments.